# Revisiting and Extending Similarity-based Metrics in Summary Factual Consistency Detection

## Abstract

Cutting-edge abstractive summarisers generate fluent summaries, but the factuality of the generated text is not guaranteed. Early summary factuality evaluation metrics are usually based on n-gram overlap and embedding similarity, but are reported fail to align with human annotations. Therefore, many techniques for detecting factual inconsistencies build pipelines around natural language inference (NLI) or question-answering (QA) models with additional supervised learning steps. In this paper, we revisit similarity-based metrics, showing that this failure stems from the use of reference texts for comparison and the granularity of the comparison. We propose a new zero-shot factuality evaluation metric, Sentence-BERT Score (SBERTScore), which compares sentences between the summary and the source document. It outperforms widely-used word-word metrics including BERTScore and can compete with existing NLI and QA-based factuality metrics on the benchmark without needing any fine-tuning. Our experiments indicate that each technique has different strengths, with SBERTScore particularly effective at identifying correct summaries. Additionally, we demonstrate how a combination of techniques is more effective at detecting various types of error. [1]

## 1 Introduction

The rapid development of natural language generation techniques has created new challenges for evaluation. For instance, in recent years, state-of-the-art abstractive summarisation models have set new records on benchmarks many times (Zhang et al., 2020; Lewis et al., 2019; Zhao et al., 2022). However, investigation (Maynez et al., 2020; Pagnoni et al., 2021; Durmus et al., 2020; Koto et al., 2022) shows that these models are prone to generate factually inconsistent summaries. Evaluation metrics, such as ROUGE (Lin, 2004), have not undergone the same pace of improvement, and therefore fail to reflect this issue. The first step towards improving summary factuality is to develop an evaluation metric to assess summary factual consistency conditioned on the given source document.

Recent factuality metrics mainly fall into two types. 1) NLI-based metrics (Kryscinski et al., 2020; Laban et al., 2022) predict the probability that the given summary is entailed by the source document. 2) QA-based metrics (Durmus et al., 2020; Fabbri et al., 2021b; Scialom et al., 2021) simulate the process of a human performing reading comprehension tasks and compute the factuality score based on how many questions generated from the summary can be correctly answered from the given source document. These two paradigms need to train their models on a large-scale dataset, but existing factuality datasets are usually insufficient.

Similarity-based metrics are proposed to handle synonyms that cause n-gram-based methods to fail (Zhang et al., 2019). However, they are not favoured by most previous work as they do not improve the performance of reflecting summary factuality (Maynez et al., 2020; Pagnoni et al., 2021; Durmus et al., 2020). In this work, our empirical experiments validate that the failure of similarity-based metrics are from unfair experimental settings. NLI and QA-based metrics compare generated summaries against source document while similarity-based metrics take a reference summary as input.

---

[1]All the code will be made available upon acceptance.

We show that BERTScore (Zhang et al., 2019) can provide useful factual consistency evaluation by comparing generated summaries to sources.

In addition, we extend the similarity-based metric to the sentence level, as comparing individual words offers very limited insights into factual consistency, and factually consistent sentences can be constructed in entirely different ways. The proposed method, Sentence-BERT Score (SBERTScore), computes cosine similarity between sentence embeddings (Reimers & Gurevych, 2019). Like BERTScore, it is computationally-efficient and can be applied out-of-the-box with no dataset-specific training, but also takes the composition and order of words into account, so can better represent the semantics of the complete sentence compared to the contextualised word embeddings used by BERTScore. We conduct a case study to show the benefits of using sentence embedding. Comparison on a factuality benchmark (Tang et al., 2023) shows that SBERTScore outperforms BERTScore in overall performance, but BERTScore can work better in the extreme case.

We also compare BERTScore and SBERTScore against recent NLI and QA-based factuality metrics. Similarity-based metrics do not require any additional training steps as they benefit from high-quality general-purpose pretrained embeddings, and has much less computational complexity at inference time. Results show that both similarity-based metrics can outperform NLI-based metrics in the same zero-shot setting, and even work better than some metrics specifically trained for factuality evaluation. Importantly, the design of BERTScore and SBERTScore avoids truncating long source documents, instead selecting an appropriate granularity to segment the sources before feeding them into the model. Further analysis of agreement between metrics, as well as the types of errors (Tang et al., 2023) they detect, indicates that SBERTScore can capture different kinds of errors than NLI and QA-based methods. We show that a simple combination of metrics can outperform the individual base metrics, which suggests that combining diverse metrics may be a promising direction for future research.

Our contributions are three-fold:

- We conduct an empirical evaluation, which reveals that the previous underperformance of metrics such as BERTScore is due to the use of reference summaries. Zero-shot similarity-based metrics are competitive with recent factuality metrics that require additional training.
- We develop the token-level similarity-based metric BERTScore into sentence-level SBERTScore and improve the performance of detecting factual inconsistency.
- We show that different evaluation metrics are necessary to capture different types of error, and introduce a simple combination that outperforms the state of the art.

## 2 RELATED WORK

### 2.1 NLI-BASED FACTUALITY METRICS

The NLI task is similar to predicting factual consistency between source document and generated summary. Hence, previous research (Barrantes et al., 2020; Falke et al., 2019) attempted to transfer NLI models to factual consistency detection. However, a subsequent study (Kryscinski et al., 2020) showed that those NLI models are only as good as random guessing. Therefore, a series of work (Kryscinski et al., 2020; Laban et al., 2022) made efforts to build up datasets for training factuality metrics. Although the dataset can be synthesised using entity swapping to save the effort of collecting human annotations, the error distribution is not the same as real summaries (Pagnoni et al., 2021). Some recent work (Utama et al., 2022; Soleimani et al., 2023) applied language generation models to augment the quality of training set and received better performance.

Another strand of research into NLI-based factuality prediction focused on the granularity of the input text. Early works (Barrantes et al., 2020; Falke et al., 2019; Kryscinski et al., 2020) concatenate the system summary with the whole source document as the input. Firstly, this often requires truncating the source document to fit the length limit, which can lead to underestimating factuality due to the information loss. Secondly, the NLI models applied in their work are trained on much shorter sentence pairs. Directly applying these models on long text such as source documents does not align with their training data distribution. Following work (Goyal & Durrett, 2020; Laban et al., 2022; Schuster et al., 2022) investigated the effect of performing inference at different levels, including word, dependency, sentence, and paragraph, revealing that segmenting source documents

into sentences and dependency arcs is more suitable for current NLI models. This inspired us to explore the suitability of different input text granularities for similarity-based evaluation metrics, which have not been investigated in past work.

## 2.2 QA-BASED FACTUALITY METRICS

QA-based metrics (Chen et al., 2018; Wang et al., 2020; Durmus et al., 2020; Fabbri et al., 2021b) assemble multiple modules with different functions. An answer selection module first selects a set of answers from the summary, usually including named entities and noun phrase chunks. A question generation module conditioned upon the selected answers is applied on the summary as context to raise questions. The QA component answers the generated questions conditioned on the given source document. The final score is then computed on the overlapping extent of the two answer sets. This paradigm provides an interpretable way to assess factuality by showing questions with inconsistent answers. However, since several text generation models are involved in the evaluation process, this methodology usually requires a large training dataset and is time-consuming at inference time. We were therefore motivated to investigate alternatives, as factuality datasets are usually small and domain-specific, and the evaluation process is expected to be prompt.

## 2.3 SIMILARITY-BASED FACTUALITY METRICS

BERTScore (Zhang et al., 2019) is used as a stronger baseline than ROUGE (Lin, 2004) in factual consistency detection, but it does not correlate well with human judgements (Pagnoni et al., 2021). Bao et al. (2023) attempted to feed source document to BERTScore, but they did not compare its performance against metrics using other methodologies. They also tried to extend BERTScore to sentence-level without using sentence embeddings, leading to unsuccessful results. Koto et al. (2022) adapted BERTScore by averaging the three highest token scores and showed that it can detect the information overlap of system summaries and source documents, but there was still a large performance gap with other metrics (Fabbri et al., 2021b). In this work, we successfully extend similarity-based metric to sentence-level and reveal that its zero-shot setting is competitive to other metrics specifically trained for factual consistency detection.

## 3 SENTENCE-BERT SCORE

BERTScore Zhang et al. (2019) computes similarity at the word-level by comparing the embeddings of words in the generated text with their closest match in the source or reference text. However, factual consistency should be judged at a higher level, as sentences containing similar words can express different meanings. Therefore, we propose the sentence-level evaluation metric, Sentence-BERT Score (SBERTScore), utilising sentence transformers (Reimers & Gurevych, 2019) to capture the meaning of the complete sentence. The *precision* and *recall* of our proposed metric are defined as follows. $S_{\{D,S\}}$ represent the sentence set of the given source document and summary respectively, and $s_{\{i,j\}}$ are the sentences in the sets.

$$SBERT_{prec} = \frac{1}{|S_S|} \sum_{s_i \in S_S} \max_{s_j \in S_D} cossim(s_i, s_j) \tag{1}$$

$$SBERT_{recall} = \frac{1}{|S_D|} \sum_{s_j \in S_D} \max_{s_i \in S_S} cossim(s_i, s_j) \tag{2}$$

In practice, sentence transformers can generate embeddings for any texts shorter than 512 tokens, which need not be single, complete sentences. Therefore, we investigate three different granularities, and test them in Section 5.3 to find the most suitable setup for SBERTScore:

**Sent** Segment the input text into sentences.

**Doc** Take the whole text as input and truncate the part that exceeds the length limit.

**Mean**    Segment the input text into sentences and take the average sentence embedding to represent the whole input.

Regarding *precision*, *recall* and *F1 measure*: precision is better suited to capturing factuality because it reflects the extent to which summary sentences are supported by source sentences. We test this hypothesis in the following Section 5.1.

### 3.1 COMPUTATIONAL EFFICIENCY

SBERTScore applies an all-purpose embedding model as the backbone, which provides reliable sentence embeddings that can be used out-of-box without the cost of additional training, in contrast to other metrics based on NLI or QA. SBERTScore also has advantages at inference time. We denote the number of sentences in the system summary and source document as $N$ and $M$ respectively. The majority of inference time is spent on calling the backbone model to process the input sentences. NLI-based metrics need to take each sentence pair once, therefore the number of inputs that the backbone model processes is $\mathcal{O}(NM)$. SBERTScore uses a similar backbone but only needs to compute the embedding once for each sentence, so the complexity is $\mathcal{O}(N + M)$. The runtime of QA-based metrics is much greater than the other two as multiple models are involved in question generation and answering, thus has the lowest efficiency. We randomly sampled 1000 pieces of data from the benchmark, and test the runtime of QuestEval (Scialom et al., 2021), SummaC$_{\{ZS,Conv\}}$ (Laban et al., 2022), BERTScore (Zhang et al., 2019) and SBERTScore on Intel(R) Core(TM) i9-10900X CPU @ 3.70GHz with NVIDIA A5000. Results in Appendix A show that SBERTScore only comes after BERTScore in processing speed, and is 3 times faster than the rival NLI-based method SummaC$_{\{ZS,Conv\}}$ and 30 times faster than the QA-based metric QuestEval.

## 4 EXPERIMENTAL SETTINGS

### 4.1 DATASETS

To evaluate our proposed factuality metric against alternatives, we use the benchmark built by Tang et al. (Tang et al., 2023), which consists of summaries and human annotations sampled from nine existing factuality datasets, including XSumFaith (XSF) (Maynez et al., 2020), Polytope (Huang et al., 2020), FactCC (Kryscinski et al., 2020), SummEval (Fabbri et al., 2021a), FRANK (Pagnoni et al., 2021), QAGS (Wang et al., 2020), CLIFF (Cao & Wang, 2021), Goyal 21' (Goyal & Durrett, 2021), and XENT (Cao et al., 2021). The dataset characteristics are shown in Table 1. All

| Dataset | Annotator Number | Size | Source Length | Summary Length |
|---------|------------------|------|---------------|----------------|
| XSF | 3 | 2353 | 505.0 | 28.1 |
| Polytope | 3 | 1268 | 691.5 | 83.1 |
| FactCC | 2 | 1434 | 728.4 | 21.8 |
| SummEval | 8 | 1698 | 453.7 | 79.2 |
| FRANK | 3 | 1393 | 692.1 | 67.5 |
| QAGS | 3 | 474 | 414.2 | 45.9 |
| CLIFF | 2 | 600 | 576.9 | 45.8 |
| Goyal' 21 | 2 | 100 | 504.3 | 29.9 |
| XENT | 5 | 696 | 436.6 | 32.9 |
| Average | 3.4 | 1112.8 | 572.8 | 50.4 |

Table 1: Dataset characteristics in the benchmark. Source/Summary Length refer to the number of tokens counted based on the results of Roberta-large tokenizer respectively(Liu et al., 2019).

source documents are English news articles, originally from the validation and test set of two news summarisation benchmarks, CNNDM (See et al., 2017) and XSum (Narayan et al., 2018). Corresponding summaries were generated by a range of abstractive summarisers, including BART (Lewis et al., 2019), PEGASUS (Zhang et al., 2020), and BERTSumAbs (Liu, 2019). We remove data from CNNDM in Goyal 21', as its validation set is extremely imbalanced (only 1 consistent example in the validation set), which impairs the classification threshold selection.

## 4.2 PERFORMANCE EVALUATION

Following previous work (Pagnoni et al., 2021; Tang et al., 2023; Laban et al., 2022; Fabbri et al., 2021b; Kryscinski et al., 2020), we select a threshold for each metric using the validation set and report their balanced accuracy and correlations to human annotation. Balanced accuracy is defined as:

$$BalancedAcc = \frac{1}{2} \left( \frac{TP}{TP + FN} + \frac{TN}{TN + FP} \right),$$

where $TP$, $TN$, $FP$, and $FN$ refer to the number of true positives, true negatives, false positives, and false negatives, respectively. In addition, we report Area Under Curve of Receiver Opearting Characteristic (ROC-AUC) (Fawcett, 2006), which does not require a threshold to reflect the metrics' ability to discriminate consistent and inconsistent summaries. Results' significance is computed via t-test.

## 4.3 EVALUATION METRICS FOR COMPARISON

This section introduces factuality metrics studied for comparison.

**QAFactEval** Fabbri et al. (2021b) conducted a comprehensive evaluation of the components of QA-based metrics. They aggregated more advanced models into a single system and optimised a pipeline for computing consistency scores.

**QuestEval** Scialom et al. (2021) proposed a QA-based framework to compute consistency scores for given text pairs. They first select an answer set from the candidate text, then generate questions using the other text as input with conditions from the answer set. The QA module answers the questions and the overlap between the two answer sets is counted to obtain precision and recall. They use F1 measure as the final factual consistency score.

**DAE** Goyal & Durrett (2020) extract dependencies from given texts using dependency parsing. They train a model to predict entailment at the dependency-level. The final score is the average entailment score over all dependency arcs in the given source and summary.

**SummaC**$_{\{ZS,Conv\}}$ Laban et al. (2022) train a sentence-level NLI model and compute the entailment scores for all pairs of sentences from the source document and the summary. **ZS** stands for zero-shot, where the final entailment score is the average of the maximum entailment score for each sentence in the summary. **Conv** is a variant with an extra learned convolutional layer that aggregates the entailment score matrix to a final score.

**ROUGE**$-\{1, 2, L\}$ Lin (2004) propose an evaluation metric by counting the overlapping words or n-grams between the given reference and candidate text pairs.

**BERTScore** Zhang et al. (2019) report the average cosine similarity of the matched word embeddings provided by BERT Devlin et al. (2018) or other related models.

FactCC, SummaC$_{\{ZS,Conv\}}$, DAE are NLI-based metrics, and QuestEval, QAFactEval are QA-based metrics. To have a fair comparison, we use the pretrained RoBERTa-large Liu et al. (2019) as the backbone for BERTScore and all-roberta-large-v1 Reimers & Gurevych (2019) for SBERTScore. The two checkpoints have identical numbers of layers, and the only difference is that they are trained for different text embeddings.

## 5 EXPERIMENTS AND RESULTS

In this section, we first investigate the suitability of different settings for similarity-based metrics. We also look into a case study to better understand the metrics' behaviour when processing negation and neutral sentences. Then we test metric performance on the benchmark. The last subsection reports the error analysis and agreement between different factuality metrics and demonstrates the benefit of metric combination.

## 5.1 Comparison of Precision, Recall, and F1

For similarity-based metrics, we compare *precision*, *recall*, and *F1 measure* to select the most informative way to measure the overlap between tokens or sentences in the source and summary. From the definition (Equation 1), *precision* relates better to the accuracy of the information included in the summary, while *recall* (Equation 2) reflects how completely the summary covers the source document. Table 2 supports our hypothesis that *precision* can assess generated summaries more accurately from the perspective of factuality. Therefore, we report *precision* of BERTScore and SBERTScore in the following sections.

| Metric | Precision | Recall | F1 |
|--------|-----------|--------|-------|
| BERTScore | 0.759 | 0.627 | 0.710 |
| SBERTScore | **0.779** | 0.644 | 0.703 |

Table 2: Average balanced accuracy on the benchmark using *precision*, *recall*, and *F1 measure*. The highest result is in **bold**, which is significantly higher than the second best result with $p < 0.05$.

## 5.2 Comparison Text Selection

We investigate the effect of taking `(source, summary)` and `(reference, summary)` as input to n-gram matching and similarity-based metrics. Table 3 shows that the choice of comparison text makes a huge difference to the same evaluation metric. The highest results on `(reference, summary)` pairs are only as good as a random guess, while the performance on `(source, summary)` pairs is greatly improved. References may be unsuitable since they carry less information than the source document, and often contain extrinsic knowledge aggregated by human writers (Maynez et al., 2020), especially in XSum (Narayan et al., 2018).

| Metric | Reference | Source |
|--------|-----------|--------|
| Rouge 1 | 0.491 | 0.638 |
| Rouge 2 | 0.318 | 0.706 |
| Rouge L | 0.491 | 0.674 |
| BERTScore | 0.500 | 0.759 |
| SBERTScore | 0.499 | **0.779** |

Table 3: Average balanced accuracy computed on different comparison texts on the benchmark. The highest result is in **bold**. All results in the source column are significantly higher than their corresponding results in the upper bracket with $p < 0.05$.

## 5.3 Text Granularity Selection

As performance can vary based on how the input text is segmented and processed before being fed into the sentence-transformer, we test the settings mentioned above in different combinations to build up a recommendation for using SBERTScore. For BERTScore, we only test word level embeddings since it has been reported that BERT does not perform well in representing higher level text embeddings (Reimers & Gurevych, 2019). For SBERTScore, we additionally test word level input to better understand the contribution of granularity to the improvement.

SBERTScore on sentence-sentence level achieves the highest score in Table 4. It also outperforms BERTScore on the same word-word level similarity, indicating that the improvement is brought by both the architecture and the appropriate text granularity. For document-level, the performance drops greatly when it is applied on the source document, as 45.76% of the source documents are truncated. Inputting the summary at document level has a much smaller effect as the summary length is usually much shorter than the length limit. Segmenting the source documents at the right granularity can avoid the information loss brought by the length limit while producing more suitable embeddings for judging factuality.

A simplification to SBERTScore is to compute the mean sentence embedding for an input document, avoiding the need to search for the maximum similarity while still processing sentences individually

| Model | Granularity | Balanced Accuracy |
|---|---|---|
| BERTScore | Word-Word | 0.759 |
| SBERTScore | Word-Word | 0.767 |
| | Sent-Sent | **0.779** |
| | Doc-Sent | 0.576 |
| | Sent-Doc | 0.746 |
| | Doc-Doc | 0.684 |
| | Mean-Sent | 0.602 |
| | Sent-Mean | 0.565 |
| | Mean-Mean | 0.512 |

Table 4: Balanced accuracy with different text granularities as input. The highest balanced accuracy is highlighted in **bold**, which is significantly higher than the second best result with $p < 0.05$.

with SBERT. In Table 4, we observe that averaging either source or summary will lead to worse balanced accuracy, which justifies the sentence granularity proposed in Section 3.

### 5.4 Case Study: Negation

BERTScore is reported to struggle at handling negation accurately (Leiter et al., 2022). Here, we present a case study to illustrate the performance of SBERTScore when processing negation. Consider the four examples sentences below:

$S_1$  I like rainy days because they make me feel relaxed

$S_2$  I don't like rainy days because they don't make me feel relaxed.

$S_3$  I enjoy rainy days because they make me feel calm.

$S_4$  I enjoy listening to music at rainy days.

Table 5 shows the BERTScores and SBERTScores obtained by comparing the given sentence pairs. BERTScore fails to identify the negation in $S_2$ and assigns a high score despite its inconsistency with $S_1$. SBERTScore does better since it works on the sentence-level where negation could have a larger influence. However, the comparison between SBERTScores of $\langle S_1, S_2 \rangle$ and $\langle S_1, S_4 \rangle$ indicates that it is not sensitive enough to distinguish between negation and neutral expressions. $\langle S_1, S_4 \rangle$ do not contradict one another, so should receive a higher score than $\langle S_1, S_2 \rangle$, yet both pairs have very similar SBERTScores. Future research is therefore required into handling negation.

| Metric | $\langle S_1, S_2 \rangle$ | $\langle S_1, S_3 \rangle$ | $\langle S_1, S_4 \rangle$ |
|---|---|---|---|
| BERTScore | 0.984 | 0.988 | 0.915 |
| SBERTScore | 0.720 | 0.975 | 0.701 |

Table 5: BERTScore and SBERTScore of example sentence pairs.

### 5.5 Benchmark Comparison with NLI and QA-based Methods

The detailed results on each sub-dataset are shown in Table 6. We find that metric performance varies across different datasets, suggesting that choosing a suitable metric will, in practice, depend on the situation. Therefore, we provide an assessment of overall performance, we combine the data from each origin (CNNDM or XSum) in Table 7a and Table 7b. For a fair comparison, we only report the DAE results on CNNDM as it is trained on human annotated XSum validation set which overlaps with the benchmark dataset.

QAFactEval outperforms other metrics on all splits of the dataset. Both similarity-based metrics outperform the zero-shot NLI baseline on both splits. SBERTScore and BERTScore achieve second best on the CNNDM and XSum split respectively, suggesting the opposite conclusion to previous studies (Fabbri et al., 2021b; Pagnoni et al., 2021; Durmus et al., 2020), that similarity-based metrics can work well and even outperform trained factuality metrics in zero-shot settings given a suitable

comparison text. SBERTScore outperforms BERTScore across the whole dataset and CNNDM split as Table 3 and Table 7 shows, but it is not as good as BERTScore on the XSum split. We speculate that this is because all summaries in the XSum split are single sentences, which highly compress the meaning from multiple sentences in the document. Therefore there is only one comparison sentence pair applied in Eq 1, preventing SBERTScore from averaging scores over sentences and leading to degeneration. Some evidence for this is that $SummaC_{ZS}$, which averages the maximum scores in each column of the score matrix in the same way as our metric, also underperforms on XSum. However, both $Summac_{Conv}$ and BERTScore, as comparable alternatives to these two metrics respectively, still average scores from several comparisons, thus having better performance.

| Metric | Dataset | | | | | | | | |
|---|---|---|---|---|---|---|---|---|---|
| | XSF | Polytope | FactCC | SEval | FRANK | QAGS | CLIFF | Goyal' 21 | XENT |
| QAFactEval | 0.604 | **0.827** | 0.843 | **0.830** | **0.729** | **0.692** | 0.703 | 0.754 | 0.613 |
| QuestEval | 0.605 | 0.708 | 0.655 | 0.713 | 0.567 | 0.607 | 0.691 | **0.797** | 0.601 |
| DAE | - | 0.782 | 0.704 | 0.716 | 0.695 | 0.586 | 0.734 | - | - |
| $SummaC_{Conv}$ | **0.655** | 0.744 | **0.891** | 0.793 | 0.655 | 0.629 | **0.744** | 0.552 | **0.668** |
| $SummaC_{ZS}$ | 0.549 | **0.786** | **0.835** | 0.781 | 0.672 | **0.673** | 0.700 | 0.466 | 0.490 |
| BERTScore | 0.527 | 0.779 | 0.632 | 0.759 | **0.676** | 0.586 | **0.724** | **0.657** | **0.601** |
| SBERTScore | **0.608** | 0.772 | 0.754 | **0.827** | 0.655 | 0.596 | 0.701 | 0.605 | 0.581 |

Table 6: Balanced accuracy of different metrics on each dataset. Metrics in the top require training while the bottom ones are zero-shot. The best results of each column in the two sections are **highlighted**. Underline indicates the result is significantly better than the second best one in the same section with $p < 0.05$.

| Metric | CNNDM | | | |
|---|---|---|---|---|
| | Balanced Acc. | ROC-AUC | Pearson $\rho$ | Spearman $\rho$ |
| QAFactEval | **0.757** | **0.823** | **0.547** | **0.469** |
| QuestEval | 0.670 | 0.736 | 0.361 | 0.343 |
| DAE | 0.696 | 0.747 | 0.405 | 0.358 |
| $SummaC_{Conv}$ | 0.737 | 0.796 | 0.446 | 0.430 |
| $SummaC_{ZS}$ | 0.686 | 0.759 | 0.407 | 0.377 |
| BERTScore | 0.692 | 0.767 | 0.405 | 0.388 |
| SBERTScore | **0.720** | **0.804** | **0.458** | **0.441** |

(a) Metric performance on the CNNDM split.

| Metric | XSum | | | |
|---|---|---|---|---|
| | Balanced Acc. | ROC-AUC | Pearson $\rho$ | Spearman $\rho$ |
| QAFactEval | **0.705** | **0.773** | **0.423** | **0.403** |
| QuestEval | 0.665 | 0.711 | 0.403 | 0.307 |
| $SummaC_{Conv}$ | 0.604 | 0.654 | 0.210 | 0.223 |
| $SummaC_{ZS}$ | 0.577 | 0.607 | 0.181 | 0.156 |
| BERTScore | **0.695** | **0.738** | **0.342** | **0.346** |
| SBERTScore | 0.605 | 0.653 | 0.227 | 0.222 |

(b) Metric performance on the XSum split.

Table 7: Performance of different metrics on each dataset split. Metrics in the top require training while the bottom ones are zero-shot. The best results of each column on the two sections are **highlighted** and are significantly better than the next best one in their section with $p < 0.05$.

### 5.6 ERROR ANALYSIS AND METRIC COMBINATION

Previous studies (Pagnoni et al., 2021; Tang et al., 2023) point out that different metrics can be sensitive to different errors, inspiring us to look into the possibility of combining different

metrics. We first investigate the error type sensitivity of BERTScore and SBERTScore, following the coarse error type taxonomy in Tang et al. (2023). Errors are classified from two perspectives. Errors made up by text pieces that appear in the source document are noted as $Intrinsic$, otherwise $Extrinsic$. The error attributes are further classified as either $NounPhrase$ or $Predicate$. All errors from XSF (Maynez et al., 2020), FRANK (Pagnoni et al., 2021), Goyal 21' (Goyal & Durrett, 2021), and CLIFF (Cao & Wang, 2021) are annotated with a subset of $\{Intrinsic, Extrinsic\} \times \{NounPhrase, Predicate\}$. For summaries from XSum, they have two special additional error types, $\{IntrinsicSentence, ExtrinsicSentence\}$, if the whole sentence is inconsistent. We report the recall of each metric in Table 8 as it reflects their sensitivity to each type of error, as well as correct summaries.

The results demonstrate that metrics have different strengths. Benefiting from the properties of similarity, BERTScore and SBERTScore perform better on extrinsic than intrinsic errors for the same attribute type. Compared to the recall of errors, the most impressive ability of SBERTScore is to identify correct summaries. It significantly outperforms all the other metrics on CNNDM, and comes only after SummaC$_{ZS}$ on XSum. High recall on correct summaries suggests that SBERTScore will not easily misjudge a consistent summary. In other words, if a summary is assigned with low SBERTScore, then it is very likely to be an unfaithful summary to the source document. We investigate the case where NLI and QA-based metrics fail but SBERTScore makes a correct judgement in Appendix B.

| Metric | CNNDM | | | | |
| | Intrinsic | | Extrinsic | | Correct |
| | NP. | P. | NP | P. | |
| --- | --- | --- | --- | --- | --- |
| QAFactEval | 0.546 | 0.509 | 0.791 | 0.633 | 0.401 |
| QuestEval | **0.695** | 0.582 | 0.777 | **0.742** | 0.309 |
| DAE | 0.575 | 0.509 | 0.668 | 0.609 | **0.436** |
| SummaC$_{Conv}$ | 0.684 | **0.782** | **0.841** | 0.711 | 0.287 |
| SummaC$_{ZS}$ | 0.632 | **0.745** | **0.800** | 0.711 | 0.314 |
| BERTScore | **0.661** | 0.636 | 0.741 | **0.719** | 0.342 |
| SBERTScore | 0.454 | 0.436 | 0.586 | 0.563 | **0.522** |

(a) Error type analysis on CNNDM.

| Metric | Xsum | | | | | | |
| | Intrinsic | | | Extrinsic | | | Correct |
| | NP. | P. | Sent. | NP | P. | Sent. | |
| --- | --- | --- | --- | --- | --- | --- | --- |
| QAFactEval | **0.671** | **0.720** | 0.882 | 0.532 | 0.631 | 0.808 | 0.304 |
| QuestEval | 0.493 | 0.553 | **0.941** | 0.520 | **0.644** | **0.849** | **0.387** |
| SummaC$_{Conv}$ | 0.551 | 0.629 | 0.294 | **0.640** | 0.619 | 0.715 | 0.371 |
| SummaC$_{ZS}$ | **0.676** | **0.652** | 0.824 | 0.569 | 0.589 | 0.523 | **0.418** |
| BERTScore | 0.538 | 0.621 | **0.882** | **0.597** | 0.631 | 0.782 | 0.375 |
| SBERTScore | 0.498 | 0.644 | 0.706 | 0.532 | **0.661** | **0.808** | 0.397 |

(b) Error type analysis on XSum.

Table 8: Recall (sensitivity) of each metric on different types of errors, as well as correct summaries. The best results of each column in the two sections are **highlighted**. Underline indicates the result is significantly better than the second best in the same section with $p < 0.05$. We remove the results of DAE for a fair comparison as it is trained on the annotated validation set of XSum.

Furthermore, we investigate the agreement among different metrics on the benchmark to find out whether they can be complementary to each other. The Kohen's $\kappa$ scores in Appendix C show weak agreement ($< 0.45$) among the metrics. Considering that these metrics have similar balanced accuracy, it suggests that a combination of comparison approaches could be more effective than relying on a single metric. We simply test this idea by combining pairs of distinct evaluation metrics using logical *AND* (both metrics must mark the summary as consistent) and *OR* (the summary is marked as consistent if at least one metric marks it consistent).

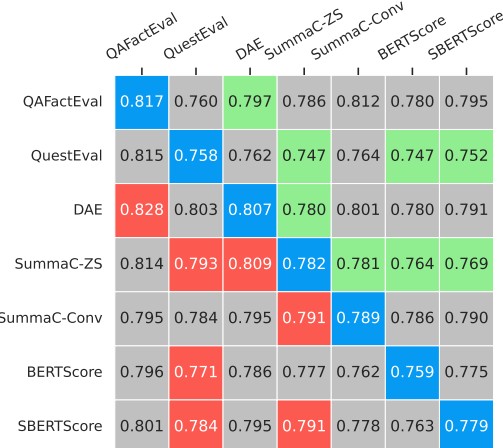

Figure 1: Average balanced accuracy of combined metrics on the benchmark. The diagonal is the balanced accuracy of the original evaluation metric (highlighted in blue). The upper triangular matrix is the balanced accuracy of joint metrics using *OR* and the lower triangular matrix is based on *AND*. Red blocks highlight the balanced accuracy that is improved over two original metrics, and green blocks highlight those are lower than both original metrics. All improvements and declines are statistically significant with $p < 0.05$.

The joint balanced accuracy of each combination is shown in Figure 1. The lower triangular matrix indicates that logical *AND* can improve the balanced accuracy, while the upper triangular matrix suggests that logical *OR* reduces performance, demonstrating that individual factuality metrics may suffer from false positives. Logical *AND* introduces a double-checking mechanism, which raises the accuracy by mitigating the false consistent rate and improving the true inconsistent rate. Appendix B shows two examples where logical operations correct a misclassification. We further investigate the use case of *AND* on SBERTScore and QuestEval and their confusion matrix in Appendix D.

## 6 CONCLUSION

In this paper, we investigated the suitable settings for similarity-based factuality evaluation metrics and proposed a new sentence-level metric, SBERTScore. We showed that, given source documents as input, similarity-based evaluation metrics computed on sentence-sentence level are competitive with more complex NLI and QA-based factuality-oriented metrics, and do not require a supervised learning step on the target domain. Furthermore, our proposed metric better aligns with human binary annotations than many trained metrics on the CNNDM split and across the whole dataset. Therefore, we conclude that zero-shot similarity-based metrics are a promising approach. We illustrate a limitation of similarity-based metrics when processing negation and highly similar but neutral input text, which suggests a direction for future research. We also showed that our proposed metric has high recall of correct summaries, and that there is low agreement between different factuality metrics, with similarity-based metrics making different errors to QA and NLI-based metrics. Building on this, we demonstrated that integrating metrics by logical *AND* can improve balanced accuracy on benchmark datasets.

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

## A    METRIC PROCESSING SPEED

We randomly sampled 1000 pieces of data from the benchmark and ran QuestEval, SummaC$_{\{ZS,Conv\}}$, BERTScore and SBERTScore on them. We did not test DAE and QAFactEval as their dependencies are not compatible with our GPU. The runtime of each metric to process 1000 pieces of data is presented in Table 9.

| Metric | Time (s) |
|---|---|
| QuestEval | 1914 |
| SummaC$_{ZS}$ | 207 |
| SummaC$_{Conv}$ | 233 |
| BERTScore | **36** |
| SBERTScore | 67 |

Table 9: The total time needed for each metric to process the 1000 pieces of samples. The fastest metric is **highlighted**.

## B    CASE STUDY

We explore the dataset and report some examples where NLI and QA-based metric fail but SBERTScore still makes correct judgements.

Marcy Smith was woken up by her son David to find their house in Glovertown, Newfoundland and Labrador, completely engulfed in flames. The whole family was able to escape, but their house is destroyed and their dog and cats did not make it. Mrs Smith said if it wasn't for her son, she and her daughter probably wouldn't have survived. David was on FaceTime to his father at the time, so was the only one awake and saw the flames out of the corner of his eye. "Within seconds of him getting us up, the flames were everywhere," Mrs Smith told the Canadian Broadcasting Corporation. "It happened so fast. We were standing in the kitchen by the wood stove and the flames just ate around me and David. The entire kitchen just disappeared while we were standing in it." She said the fire was started by some rubbish she burned in the wood stove, something she had done "a thousand times" before. The fire alarm did not go off. The family had nothing but pyjamas on when they fled, but Mrs Smith said the community has rallied behind them, donating clothes and shoes and even a bike for her son. "All he understands is that me and his sister and him got out. He does not understand that he is the only reason we did," she said. "He did a huge thing for such a young boy. I am so proud of him and I am going to tell him for the rest of his life until he understands what a big thing he did."

A Canadian family who survived a house fire has been reunited with their family.

(a) Example 1: Inconsistent text is marked in red.

The Tulsa County reserve deputy who fatally shot a man instead of using his Taser turned himself in to authorities Tuesday at the Tulsa County Jail. Video shows Reserve Deputy Robert Bates announcing he is going to deploy his Taser after an undercover weapons sting on April 2 but then shooting Eric Courtney Harris in the back with a handgun. Bates was charged with second-degree manslaughter Monday. He surrendered Tuesday morning, accompanied by his attorney... Harris' brother, Andre Harris, told CNN that he is pleased District Attorney Steve Kunzweiler pressed charges. In his opinion, however, no type of force should have been used in the arrest of his brother. Watching the video of the shooting, Andre Harris said he can see that three or more officers were already on top of his brother. That manpower should have been enough to arrest him, he said... The family said that the sheriff has not apologized and that the department has not shown remorse or indication it will change its policies. CNN's Jason Morris and Ed Lavandera contributed to this report.

Eric Harris' brother says no type of force should have been used. Robert Bates is charged with second - degree manslaughter.

(b) Example 2: Evidence is marked in blue.

Table 10: Examples from the benchmark dataset where SBERTScore makes correct judgment while QuestEval and SummaC$_{Conv}$ misclassify the summary.

| Metric | Example 1 | | Example 2 | |
|---|---|---|---|---|
| | Score | Label | Score | Label |
| QuestEval | 0.397 | 1 | 0.481 | 0 |
| SummaC$_{Conv}$ | 0.293 | 1 | 0.264 | 0 |
| SBERTScore | 0.565 | 0 | 0.811 | 1 |
| GroundTruth | - | 0 | - | 1 |

Table 11: Metric results on the two examples.

Table 10 presents two stories from the benchmark dataset, and Table 11 shows their scores and labels under QuestEval, SummaC$_{Conv}$, and SBERTScore. In the first example, every noun phrase is mentioned in the source document, which could be responsible for the misclassification of QuestEval and SummaC$_{Conv}$. However, there is no source sentence mentioning "reunit" so it is assigned a low SBERTScore. Regarding the second example, we speculate that too many people and named entities are mentioned, so they, along with coreferences, could confuse QuestEval because it judges factual consistency on top of them. On the other hand, the evidence for the first summary sentence is dispersed in multiple sentences, causing difficulties for SummaC$_{Conv}$ to find it. These evidence sentences, however, are similar to the summary sentence in some extent, which results in a high SBERTScore, thus correct judgement.

## C  INTER-METRIC AGREEMENT

We compute Cohen's $\kappa$ among all metrics using their binary predictions on the benchmark. Figure 2 shows the agreement between the metrics.

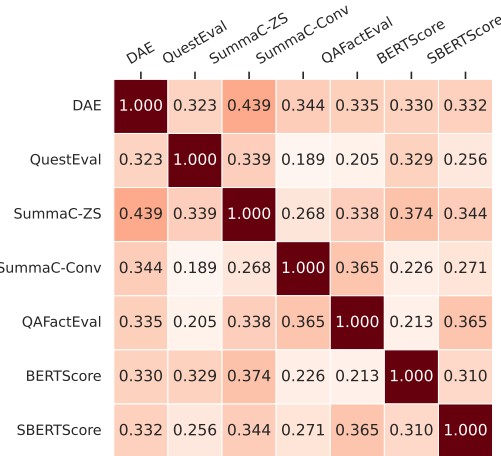

Figure 2: Cohen's $\kappa$ agreement score among different metrics on the benchmark dataset. The higher agreement is in deeper red.

## D  METRIC COMBINATION

Appendix B shows two examples where *AND* and *Or* can fix the misclassification. We look into the confusion matrices of the base metrics and the *AND* combination, as shown below in Table 12. It supports our intuition that combination can mitigate false consistent (false positive, FP) and improve true inconsistent (true negative, TN) rates, leading to a better overall performance.

| Metric | TP | TN | FP | FN | Balanced Acc. |
|---|---|---|---|---|---|
| SBERTScore | 0.444 | 0.332 | 0.084 | 0.141 | 0.779 |
| QuestEval | 0.511 | 0.266 | 0.150 | 0.074 | 0.758 |
| Combined | 0.418 | 0.355↑ | 0.061↓ | 0.166 | 0.784↑ |

Table 12: Confusion matrices of different metrics and their combined metric on the benchmark.