# OpenReview forum: "Revisiting and Extending Similarity-based Metrics in Summary Factual Consistency Detection"
_ICLR.cc/2025/Conference — Submitted to ICLR 2025_

### Official Review · Reviewer_HCzy · 2024-10-18

**Soundness:** 2
**Presentation:** 2
**Contribution:** 2
**Rating:** 3
**Confidence:** 4

**Summary:**

This paper introduces and evaluates a new metric, SBERTScore for assessing the factual consistency of abstractive summaries. SBERTScore is a sentence-level bertscore. The authors highlight that the reason that previous BERTScore, often fails than other metrics in this task is due to (1) relying on reference texts, and (2) focus on similarity at a word level. Thus, SBERTScore uses sentence embeddings and directly compares the generated summary to the source document at sentence level. SBERTScore outperforms BERTScore and competes with other NLI-based and QA-based metrics in Aggrefact. Additionally, the author found combining different metrics can improve detection of diverse types of factual errors.

**Strengths:**

* The paper is well-structured, with clear and objective writing,

*  The authors thoroughly explore various settings for using sentence-level BERTScore in the context of summary factual consistency detection,

* The idea of combining different metrics  is promising

**Weaknesses:**

* Using sentence embedding of paired texts to assess their semantic similarity is not entirely novel. Similar approaches have employed in various nlp tasks such as machine translation, paraphrasing, and also summarisation. The key difference lies in the context of comparison: when comparing generated text with a reference, it evaluates informativeness, whereas when comparing with the source document, it evaluates faithfulness.
* The size of the models used in the comparison are inconsistent, potentially affecting the fairness of the evaluation. For instance, SummaC utilizes DeBERTaV3-large (approximately the size of bert-base), QA metrics use T5-large, while BERTScore uses RoBERTa-large.
* It is unclear whether the authors used the default settings of the summac package or implemented a custom version. Summac package uses an entailment-minus-contradiction score for zs and entailment score for the conv. In fact, using only the entailment score for both variants can lead to better performance. I obtained over 70 ROC-AUC on the XSum split of Aggrefact using summac zs.

**Questions:**

* Which package did you use to segment the document into sentences?

* Section 5.4 is particularly interesting. NLI models are also sometimes fooled by similar lexical overlap. For example, NLI model may think the following premise entails the hypothesis:

Premise: "The actor was encouraged by the lawyer."

Hypothesis: "The actor encouraged the lawyer."

Did you observe similar trends with SBERTScore, where lexical overlap causes misjudgments?

* For Tables 3 and 4, were the experiments conducted on the validation set or the test set of Aggrefact?

* For Table 4, the results suggest that segmenting documents and summaries into sentences yields the best performance, while the "mean method" (i.e., averaging the embeddings of sentences) leads to worse performance. Does this conclusion hold for other datasets as well, or is it specific to AggreFact?

---

### Official Review · Reviewer_HCQS · 2024-11-03

**Soundness:** 2
**Presentation:** 3
**Contribution:** 2
**Rating:** 5
**Confidence:** 4

**Summary:**

This paper revisits the efficacy of BERTScore and SBERTScore for evaluating summary factual consistency. Specifically, it demonstrates that if the text used for comparison is changed from reference summaries to source documents, their accuracy will  substantially increase. Then, the authors also show that SBERTScore on sentence-sentence level outperforms metrics of other granularity settings. Moreover, experimental results exhibit that BERTScore and SBERTScore achieve the second best accuracy, always worse than NLI-based metrics or QA-based metrics. Finally, the paper discovers that using AND to combine the results of two metrics is more likely to obtain a higher accuracy than relying on a single metric.

**Strengths:**

- This paper discusses a potential misuse of similarity-based evaluation metrics for evaluating summary faithfulness. This is beneficial to the community as many researchers are aware of this.
- As pointed out in Section 3.1, similarity-based metrics are highly efficent than other types of metrics. The efficiency analysis is meaningful and rarely seen in prior studies.

**Weaknesses:**

- Although it is great to show that the performance of BERTScore and SBERTScore goes up a lot after changing reference texts to source documents, they still lag behind QA-based metrics or NLI-based metrics. Moreover, some newer and better evaluation methods are not compared in this study, such as AlignScore[1], AMRFact[2], and LLM-based evaluation metrics. Considering the fact that QA-based metrics and NLI-based metrics are already suboptimal, the advantage of similarity-based metrics is only from efficiency.
- It seems hard for this paper to balance two goals: emphasizing the advantages of similarity-based metrics and re-evaluating automatic evaluation metrics for summary faithfulness. For example, Section 5.4 fully belongs to the former while Section 5.6 especially Figure 1 almost corresponds to the latter.

[1] [AlignScore: Evaluating Factual Consistency with A Unified Alignment Function](https://aclanthology.org/2023.acl-long.634) (Zha et al., ACL 2023)

[2] [AMRFact: Enhancing Summarization Factuality Evaluation with AMR-Driven Negative Samples Generation](https://aclanthology.org/2024.naacl-long.33) (Qiu et al., NAACL 2024)

**Questions:**

- A missing dot in line 347

- As mentioned in Weakness, I would suggest the authors focus on one objective. If the aim is to propose a similarity-based metric, it may be better to further improve the efficacy. Besides, other automatic evaluation metrics (especially the latest ones) for summary faithfulness should be considered.

---

### Official Review · Reviewer_HxTz · 2024-11-03

**Soundness:** 3
**Presentation:** 3
**Contribution:** 2
**Rating:** 3
**Confidence:** 3

**Summary:**

Authors propose SBERTScore, a zero-shot, similarity-based metric using sentence-level embeddings for evaluating factual consistency in summarization. The authors show that token-level similarity-based metrics, such as BERTScore, have inadequate granularity for comparing factuality. Therefore they propose to compare summary-source sentence embeddings for evaluating consistency. Empirical results demonstrate that SBERTScore outperforms BERTScore and also competes with established metrics like NLI and QA-based models without additional training.

**Strengths:**

Authors discuss limitations of token-level similarity based BERTScore in evaluating factuality and proposes SBERTScore that compares sentence-level embedding and shows superior performance.

SBERTScore is efficient as it requires calculating sentence embedding once and is faster than NLI- or QA-based metrics.

**Weaknesses:**

The method may seem somewhat outdated given the field's shift towards using LLMs as general purpose evaluator for factual consistency.

It is unclear what aspect of factuality does the SBERTScore capture better than other metrics. While the results suggest SBERTScore has some strengths, it is ambiguous exactly where and why we should use it?

**Questions:**

- SBERTScore neither on its own, nor in combination with other metrics, give the best performance. For instance, if we look at Figure 1, QAFactEval alone performs better than SBERTScore  + any other metric. Also, there has been a recent shift towards LLM Judge. What do you see as the practical applications of SBERTScore?

---

### Official Review · Reviewer_2JH5 · 2024-11-05

**Soundness:** 2
**Presentation:** 3
**Contribution:** 2
**Rating:** 3
**Confidence:** 4

**Summary:**

This work proposes to use a general-purpose SBERT (SentenceBert) to evaluate factual consistency in summarization, which does not require task-specific training, and is computationally efficient compared to existing approaches. Specifically, SBERT is firstly utilized to encode the summary and document; different encoding granularities are explored, such as encoding by sentences, by document, or mean pooling. After obtaining the embeddings, cosine-similarity is directly computed between the summary and document as the factuality evaluation. Experiments on AggreFact benchmark (Tang et al., 2023) suggest that the proposed approach outperforms previous NLI-based metrics, though still lags behind previous state-of-the-art.

**Strengths:**

- The method proposed in this work is simple and straightforward, leveraging existing SBERT for the encoding; cosine-similarity computation is also lightweight. Especially, the proposed method surpasses the vanilla NLI-based baseline.

- Analysis is conducted to examine the best usage of SBERT (e.g. best encoding granularity) along with its advantages over baselines.

**Weaknesses:**

- Though the proposed method is simple and proven effective, the significance is limited due to the performance gap behind QA-based metrics (and could be also behind other strong NLI-based metrics not included in the experiments). In addition, the performance of the proposed approach is capped by the quality of pretrained SBERT. If SBERT is not largely improved, the proposed method then has little room for improvement.

- Only the vanilla NLI-based metric is adopted as the baseline. There are multiple previous NLI-based metrics that are excluded in the evaluation, which have strong performance on the same AggreFact benchmark.

  - Yin et al. (ACL 2021). DocNLI: A large-scale dataset for document-
    level natural language inference.

  - Utama et al. (NAACL 2022). Falsesum: Generating
    document-level NLI examples for recognizing factual inconsistency in summarization.

  - Zha et al. (ACL 2023). AlignScore: Evaluating factual consistency
    with a unified alignment function.

  - Zha et al. (NeurIPS 2023). Text alignment is an efficient unified model
    for massive NLP tasks.

  - Qiu et al. (ACL 2024). Amrfact: Enhancing summarization factuality evaluation with amr-driven negative samples generation

- It would be more complete to include LLM-based metrics for performance comparison and analysis, as there are many recent works focusing on utilizing LLMs for factuality detection in summarization.

  - Liu et al. (EMNLP 2023). G-eval:
    NLG evaluation using gpt-4 with better human alignment.

  - Chen et al. (2023). Evaluating Factual Consistency of Summaries with Large Language Models.

  - Wu et al. (2023). Less is More for Long Document Summary Evaluation by LLMs.

  - Xu et al. (EMNLP 2024). Identifying Factual Inconsistencies in Summaries:
    Grounding Model Inference via Task Taxonomy.

**Questions:**

The reported baseline numbers seem different from those in the original AggreFact paper (Tang et al., 2023). How is the balanced accuracy computed for CNNDM and XSUM?

---

### Meta-Review · Area_Chair_dgVz · 2024-12-19

**Metareview:**

The paper introduces SBERT, a sentence-level variant of BERTScore, to evaluate the factuality of generated summaries. Reviewers agree that the paper is missing some important baselines (e.g. NLI metrics like AlignScore, FalseSum, others like AMRFact, etc.) The experiments are conducted on the Aggrefact dataset which only includes summaries from pre-GPT3 language models. The paper should include additional experiments -- more baselines and datasets containing summaries from recent models -- to demonstrate the strength their approach.

**Additional Comments On Reviewer Discussion:**

No rebuttal posted

---

### Decision · Program_Chairs · 2025-01-22

Reject